# LATENT NORMALIZING FLOWS FOR MANY-TO-MANY CROSS-DOMAIN MAPPINGS

**Shweta Mahajan, Iryna Gurevych, Stefan Roth**
Department of Computer Science, TU Darmstadt, Germany

## ABSTRACT

Learned joint representations of images and text form the backbone of several important cross-domain tasks such as image captioning. Prior work mostly maps both domains into a common latent representation in a purely supervised fashion. This is rather restrictive, however, as the two domains follow distinct generative processes. Therefore, we propose a novel semi-supervised framework, which models shared information between domains and domain-specific information separately. The information shared between the domains is aligned with an invertible neural network. Our model integrates normalizing flow-based priors for the domain-specific information, which allows us to learn diverse many-to-many mappings between the two domains. We demonstrate the effectiveness of our model on diverse tasks, including image captioning and text-to-image synthesis.

## 1 INTRODUCTION

Joint image-text representations find application in cross-domain tasks such as image-conditioned text generation (captioning; Mao et al., 2015; Karpathy & Fei-Fei, 2017; Xu et al., 2018) and text-conditioned image synthesis (Reed et al., 2016). Yet, image and text distributions follow distinct generative processes, making joint generative modeling of the two distributions challenging.

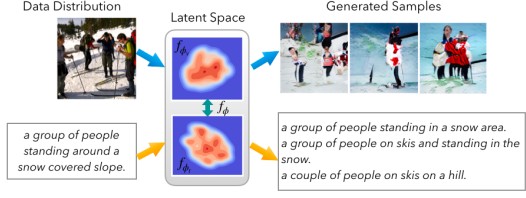

Figure 1. Joint multimodal latent representation of images and texts of our LNFMM model for diverse many-to-many mappings

Current state-of-the-art models for learning joint image-text distributions encode the two domains in a common shared latent space in a fully supervised setup (Gu et al., 2018; Wang et al., 2019). While such approaches can model supervised information in the shared latent space, they do not preserve domain-specific information. However, as the domains under consideration, *e.g.* images and texts, follow distinct generative processes, many-to-many mappings naturally emerge – there are many likely captions for a given image and vice versa. Therefore, it is crucial to also encode domain-specific variations in the latent space to enable many-to-many mappings.

State-of-the-art models for cross-domain synthesis leverage conditional variational autoencoders (VAEs, cVAEs; Kingma & Welling, 2014) or generative adversarial networks (GANs; Goodfellow et al., 2014) for learning conditional distributions. However, such generative models (*e.g.*, Wang et al., 2017; Aneja et al., 2019) enforce a Gaussian prior in the latent space. Gaussian priors can result in strong regularization or posterior collapse as they impose stringent constraints while modeling complex distributions in the latent space (Tomczak & Welling, 2018). This severely limits the accuracy and diversity of the cross-domain generative model.

Recent work (Ziegler & Rush, 2019; Bhattacharyya et al., 2019) has found normalizing flows (Dinh et al., 2015) advantageous for modeling complex distributions in the latent space. Normalizing flows can capture a high degree of multimodality in the latent space through a series of transformations from a simple distribution to a complex data-dependent prior. Ziegler & Rush (2019) apply normalizing flow-based priors in the latent space of unconditional variational autoencoders for discrete distributions and character-level modeling.

We propose to leverage normalizing flows to overcome the limitations of existing cross-domain generative models in capturing heterogeneous distributions and introduce a novel semi-supervised *Latent Normalizing Flows for Many-to-Many Mappings (LNFMM)* framework. We exploit normalizing flows (Dinh et al., 2015) to capture complex joint distributions in the latent space of our model (Fig. 1). Moreover, since the domains under consideration, *e.g.* images and texts, have different generative processes, the latent representation for each distribution is modeled such that it contains both shared cross-domain information as well as domain-specific information. The latent dimensions constrained by supervised information from paired data model the common (semantic) information across images and texts. The diversity within the image and text distributions, *e.g.* different visual or textual styles, are encoded in the residual latent dimensions, thus preserving domain-specific variation. We can hence synthesize diverse samples from a distribution given a reference point in the other domain in a many-to-many setup. We show the benefits of our learned many-to-many latent spaces for real-world image captioning and text-to-image synthesis tasks on the COCO dataset (Lin et al., 2014). Our model outperforms the current state of the art for image captioning *w.r.t.* the Bleu and CIDEr metrics for accuracy as well as on various diversity metrics. Additionally, we also show improvements in diversity metrics over the state of the art in text-to-image generation.

## 2 RELATED WORK

**Diverse image captioning.** Recent work on image captioning introduces stochastic behavior in captioning and thus encourages diversity by mapping an image to many captions. Vijayakumar et al. (2018) sample captions from a very high-dimensional space based on word-to-word Hamming distance and parts-of-speech information, respectively. To overcome the limitation of sampling from a high-dimensional space, Shetty et al. (2017); Dai et al. (2017); Li et al. (2018) build on Generative Adversarial Networks (GANs) and modify the training objective of the generator, matching generated captions to human captions. While GAN-based models can generate diverse captions by sampling from a noise distribution, they suffer on accuracy due to the inability of the model to capture the true underlying distribution. Wang et al. (2017); Aneja et al. (2019), therefore, leverage conditional Variational Autoencoders (cVAEs) to learn latent representations conditioned on images based on supervised information and sequential latent spaces, respectively, to improve accuracy and diversity. Without supervision, cVAEs with conditional Gaussian priors suffer from posterior collapse. This results in a strong trade-off between accuracy and diversity; *e.g.* Aneja et al. (2019) learn sequential latent spaces with a Gaussian prior to improve diversity, but suffer on perceptual metrics. Moreover, sampling captions based only on supervised information limits the diversity in the captions. In this work we show that by learning complex multimodal priors, we can model text distributions efficiently in the latent space without specific supervised clustering information and generate captions that are more diverse *and* accurate.

**Diverse text-to-image synthesis.** State-of-the-art methods for text-to-image synthesis are based on conditional GANs (Reed et al., 2016). Much of the research for text-conditioned image generation has focused on generating high-resolution images similar to the ground truth. Zhang et al. (2017; 2019b) introduce a series of generators in different stages for high-resolution images. AttnGAN (Xu et al., 2018) and MirrorGAN (Qiao et al., 2019) aim at synthesizing fine-grained image features by attending to different words in the text description. Dash et al. (2017) condition image generation on class information in addition to texts. Yin et al. (2019) use a Siamese architecture to generate images with similar high-level semantics but different low-level semantics.In this work, we instead focus on generating diverse images for a given text with powerful latent semantic spaces, unlike GANs with Gaussian priors, which fail to capture the true underlying distributions and result in mode collapse.

**Normalizing flows & Variational Autoencoders.** Normalizing flows (NF) are a class of density estimation methods that allow exact inference by transforming a complex distribution to a simple distribution using the change-of-variables rule. Dinh et al. (2015) develop flow-based generative models with affine transformations to make the computation of the Jacobian efficient. Recent works (Dinh et al., 2017; Kingma & Dhariwal, 2018; Ardizzone et al., 2019; Behrmann et al., 2019) extend flow-based generative models to multi-scale architectures to model complex dependencies across dimensions. Vanilla Variational Autoencoders (VAEs; Kingma & Welling, 2014) consider simple Gaussian priors in the latent space. Simple priors can provide very strong constraints, resulting in poor latent representations (Hoffman & Johnson, 2016). Recent work has, therefore, considered modeling complex priors in VAEs. Particularly, Wang et al. (2017); Tomczak & Welling (2018)

propose mixtures of Gaussians with predefined clusters, Chen et al. (2017) use neural autoregressive model priors, and van den Oord et al. (2017) use discrete models in the latent space, which improves results for image synthesis. Ziegler & Rush (2019) learn a prior based on normalizing flows to model multimodal discrete distributions of character-level texts in the latent spaces with nonlinear flow layers. However, this invertible layer is difficult to be optimized in both directions. Bhattacharyya et al. (2019) learn conditional priors based on normalizing flows to model conditional distributions in the latent space of cVAEs. In this work, we learn a conditional prior using normalizing flows in the latent space of our variational inference model, modeling joint complex distributions in the latent space, particularly of images and texts for diverse cross-domain many-to-many mappings.

## 3  METHOD

To learn joint distributions $p_\mu(x_t, x_v)$ of texts and images that follow distinct generative processes with ground-truth distributions $p_t(x_t)$ and $p_v(x_v)$, respectively, in a semi-supervised setting, we formulate a novel joint generative model based on variational inference: *Latent Normalizing Flows for Many-to-Many Mappings* (LNFMM). Our model defines a joint probability distribution over the data $\{x_t, x_v\}$ and latent variables $z$ with a distribution $p_\mu(x_t, x_v, z) = p_\mu(x_t, x_v|z)p_\mu(z)$, parameterized by $\mu$. We maximize the likelihood of $p_\mu(x_t, x_v)$ using a variational posterior $q_\theta(z|x_t, x_v)$, parameterized by variables $\theta$. As we are interested in jointly modeling distributions with distinct generative processes, *e.g.* images and texts, the choice of the latent distribution is crucial. Mapping to a shared latent distribution can be very restrictive (Xu et al., 2018). We begin with a discussion of our variational posterior $q_\theta(z|x_t, x_v)$ and its the factorization in our LNFMM model, followed by our normalizing flow-based priors, which enable $q_\theta(z|x_t, x_v)$ to be complex and multimodal, allowing for diverse many-to-many mappings.

**Factorizing the latent posterior.** We choose a novel factorized posterior distribution with both shared and domain-specific components. The shared component $z_s$ is learned with supervision from paired image-text data and encodes information common to both domains. The domain-specific components encode information that is unique to each domain, thus preserving the heterogeneous structure of the data in the latent space. Specifically, consider $z_t$ and $z_v$ as the latent variables to model text and image distributions. Recall from above that $z_s$ denotes the shared latent variable for supervised learning, which encodes information shared between the data points $x_t$ and $x_v$. Given this supervised information, the residual information specific to each domain is encoded in $z'_t$ and $z'_v$. This leads to the factorization of the variational posterior of our LNFMM model with $z_t = [z_s \ z'_t]$ and $z_v = [z_s \ z'_v]$,

$$\log q_\theta(z_s, z'_t, z'_v|x_t, x_v) = \log q_{\theta_1}(z_s|x_t, x_v) + \log q_{\theta_2}(z'_t|x_t, z_s) + \log q_{\theta_3}(z'_v|x_v, z_s). \tag{1}$$

Next, we derive our LNFMM model in detail. Since directly maximizing the log-likelihood of $p_\mu(x_t, x_v)$ with the variational posterior is intractable, we derive the log-evidence lower bound for learning the posterior distributions of the latent variables $z = \{z_s, z'_t, z'_v\}$.

### 3.1  DERIVING THE LOG-EVIDENCE LOWER BOUND

Maximizing the marginal likelihood $p_\mu(x_t, x_v)$ given a set of observation points $\{x_t, x_v\}$ is generally intractable. Therefore, we develop a variational inference framework that maximizes a variational lower bound on the data log-likelihood – the log-evidence lower bound (ELBO) with the proposed factorization in Eq. (1),

$$\log p_\mu(x_t, x_v) \geq \mathbb{E}_{q_\theta(z|x_t, x_v)}\big[\log p_\mu(x_t, x_v|z)\big] + \mathbb{E}_{q_\theta(z|x_t, x_v)}\big[\log p_\phi(z) - \log q_\theta(z|x_t, x_v)\big], \tag{2}$$

where $z = \{z_s, z'_t, z'_v\}$ are the latent variables. The first expectation term is the reconstruction error. The second expectation term minimizes the KL-divergence between the variational posterior $q_\theta(z|x_t, x_v)$ and a prior $p_\phi(z)$. Taking into account the factorization in Eq. (1), we now derive the ELBO for our LNFMM model. We first rewrite the reconstruction term as

$$\mathbb{E}_{q_\theta(z_s, z'_t, z'_v|x_t, x_v)}\big[\log p_\mu(x_t|z_s, z'_t, z'_v) + \log p_\mu(x_v|z_s, z'_t, z'_v)\big], \tag{3}$$

which assumes conditional independence given the domain-specific latent dimensions $z_t'$, $z_v'$ and the shared latent dimensions $z_s$. Thus, the reconstruction term can be further simplified as

$$\mathbb{E}_{q_{\theta_1}(z_s|x_t,x_v)q_{\theta_2}(z_t'|x_t,z_s)}\big[\log p_\mu(x_t|z_s,z_t')\big] + \mathbb{E}_{q_{\theta_1}(z_s|x_t,x_v)q_{\theta_3}(z_v'|x_v,z_s)}\big[\log p_\mu(x_v|z_s,z_v')\big]. \quad (4)$$

Next, we simplify the KL-divergence term on the right of Eq. (2). We use the chain rule along with Eq. (1) to obtain

$$D_{\text{KL}}\big(q_\theta(z_s,z_t',z_v'|x_t,x_v)\,\|\,p_\phi(z_s,z_t',z_v')\big) = D_{\text{KL}}\big(q_{\theta_1}(z_s|x_t,x_v)\,\|\,p_{\phi_s}(z_s)\big)+ \quad (5)$$
$$D_{\text{KL}}\big(q_{\theta_2}(z_t'|x_t,z_s)\,\|\,p_{\phi_t}(z_t'|z_s)\big) + D_{\text{KL}}\big(q_{\theta_3}(z_v'|x_v,z_s)\,\|\,p_{\phi_v}(z_v'|z_s)\big).$$

This assumes a factorized prior of the form $p_\phi(z_s,z_t',z_v') = p_{\phi_s}(z_s)p_{\phi_t}(z_t'|z_s)p_{\phi_v}(z_v'|z_s)$, consistent with our conditional independence assumptions, given that information specific to each distribution is encoded in $\{z_t', z_v'\}$. The final ELBO can then be expressed as

$$\log p_\mu(x_t, x_v) \geq \mathbb{E}_{q_{\theta_1}(z_s|x_t,x_v)q_{\theta_2}(z_t'|x_t,z_s)}\big[\log p_\mu(x_t|z_s,z_t')\big]$$
$$+\mathbb{E}_{q_{\theta_1}(z_s|x_t,x_v)q_{\theta_3}(z_v'|x_v,z_s)}\big[\log p_\mu(x_v|z_s,z_v')\big] - D_{\text{KL}}\big(q_{\theta_1}(z_s|x_t,x_v)\,\|\,p_{\phi_s}(z_s)\big) \quad (6)$$
$$-D_{\text{KL}}\big(q_{\theta_2}(z_t'|x_t,z_s)\,\|\,p_{\phi_t}(z_t'|z_s)\big) - D_{\text{KL}}\big(q_{\theta_3}(z_v'|x_v,z_s)\,\|\,p_{\phi_v}(z_v'|z_s)\big).$$

In the standard VAE formulation (Kingma & Welling, 2014), the priors corresponding to $p_{\phi_t}(z_t'|z_s)$ and $p_{\phi_v}(z_v'|z_s)$ are modeled as standard normal distributions. However, Gaussian priors limit the expressiveness of the model in the latent space since they result in strong constraints on the posterior (Tomczak & Welling, 2018; Razavi et al., 2019; Ziegler & Rush, 2019). Specifically, optimizing with a Gaussian prior pushes the posterior distribution towards the mean, limiting diversity and hence generative power (Tomczak & Welling, 2018). This is especially true for complex multimodal image and text distributions. Furthermore, alternatives like Gaussian mixture model-based priors (Wang et al., 2017) also suffer from similar drawbacks and additionally depend on predefined heuristics like the number of components in the mixture model. Analogously, the VampPrior (Tomczak & Welling, 2018) depends on a predefined number of pseudo-inputs to learn the prior in the latent space. Similar to Ziegler & Rush (2019); Bhattacharyya et al. (2019), which learn priors based on exact inference models, we propose to learn the conditional priors $p_{\phi_t}(z_t'|z_s)$ and $p_{\phi_v}(z_v'|z_s)$ jointly with the variational posterior in Eq. (1) using normalizing flows.

### 3.2 Variational Inference with Normalizing Flow-based Priors

Normalizing flows are exact inference models, which can map simple distributions to complex densities through a series of $K_t$ invertible mappings,

$$f_{\phi_t} = f_{\phi_t}^{K_t} \circ f_{\phi_t}^{K_t-1} \circ \cdots \circ f_{\phi_t}^1.$$

This allows us to transform a simple base density $\epsilon \sim p(\epsilon)$ to a complex multimodal conditional prior $p_{\phi_t}(z_t'|z_s)$ (and correspondingly to $p_{\phi_v}(z_v'|z_s)$). The likelihood of the latent variables under the base density can be obtained using the change-of-variables formula. A composition of invertible mappings $f_{\phi_t}^i$, parameterized by parameters $\phi_t$, is learned such that $\epsilon = f_{\phi_t}^{-1}(z_t'|z_s)$. The log-likelihood with Jacobian $J_{\phi_t}^i = \partial f_{\phi_t}^i/\partial f_{\phi_t}^{i-1}$, assuming $f_{\phi_t}^0 = I$ is the identity, can be expressed as

$$\log p_{\phi_t}(z_t'|z_s) = \log p\big(f_{\phi_t}^{-1}(z_t'|z_s)\big) - \log\left|\det\frac{\partial z_t'}{\partial\epsilon}\right|$$
$$= \log p\big(f_{\phi_t}^{-1}(z_t'|z_s)\big) - \sum_{i=1}^{K_t}\log\big|\det J_{\phi_t}^i\big|. \quad (7)$$

Using data-dependent and non-volume preserving transformations, multimodal priors can be jointly learned in the latent space, allowing for more complex posteriors and better solutions of the evidence lower bound. Using Eq. (7), the ELBO with normalizing flow-based priors can be expressed by rewriting the KL-divergence terms in Eq. (6) as (analogously for the other term)

$$D_{\text{KL}}\big(q_{\theta_2}(z_t'|x_t,z_s)\,\|\,p_{\phi_t}(z_t'|z_s)\big) = - \mathbb{E}_{q_{\theta_2}(z_t'|x_t,z_s)}\Big[\log p\big(f_{\phi_t}^{-1}(z_t'|z_s)\big) - \sum_{i=1}^{K_t}\log\big|\det J_{\phi_t}^i\big|\Big]$$
$$+ \mathbb{E}_{q_{\theta_2}(z_t'|x_t,z_s)}\big[\log q_{\theta_2}(z_t'|x_t,z_s)\big]. \quad (8)$$

Next, we describe our complete model for learning joint distributions with latent normalizing flows using Eqs. (6) and (8), which enables many-to-many mappings between domains.

## 3.3 LATENT NORMALIZING FLOW MODEL FOR MANY-TO-MANY MAPPINGS

We illustrate our complete model in Fig. 2. It consists of two domain-specific encoders to learn the domain-specific latent posterior distributions $q_{\theta_2}(z_t'|x_t, z_s)$ and $q_{\theta_3}(z_v'|x_t, z_s)$. As the shared latent variable $z_s$ encodes information common to both domains, it holds that $q_{\theta_1}(z_s|x_t, x_v) = q_{\theta_1}(z_s|x_t) = q_{\theta_1}(z_s|x_v)$ for a matching pair of data points $(x_t, x_v)$. Therefore, each encoder must be able to model the common supervised information independently for every matching pair $(x_t, x_v)$. We enforce this by splitting the output dimensions of each encoder into $z_v = [z_s \ z_v']$ and $z_t = [z_s \ z_t']$, respectively (cf. Eq. 1), and constraining the supervised latent dimensions to encode the same information. We assume that the shared latent code $z_s$ has $d'$ dimensions. We propose to learn the posterior distribution $q_{\theta_1}(z_s|x_t, x_v)$ as the shared latent space between two domain-specific au-

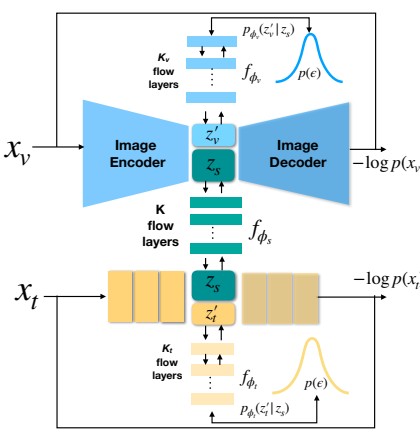

Figure 2. Our LNFMM architecture

toencoders. One simple method to induce sharing is by minimizing the mean-squared error between the encodings. However, this is not ideal given that $x_t$ and $x_v$ follow different highly multimodal generative processes. We, therefore, learn an invertible mapping $f_{\phi_s} : \mathbb{R}^{d'} \to \mathbb{R}^{d'}$ with invertible neural networks such that the $d'$-dimensional latent code $z_s$ can be transformed between the domains $v$ and $t$. Let $z_v$ with $d \geq d'$ dimensions be the encoded latent variable for distribution $p_v(x_v)$. A bijective mapping $f_{\phi_s} : (z_v)_{d'} \mapsto (z_t)_{d'}$ is learned with an invertible mapping analogous to Eq. (7) as

$$\log q_{\theta_1}(z_s|x_t, x_v) = \log q_{\theta_1}\big((z_v)_{d'}|x_t, x_v\big) = \log q_{\theta_1}\big(f_{\phi_s}((z_v)_{d'})|x_t\big) + \sum_{i=1}^{K} \log \big|\det J_{\phi_s}^i\big|. \quad (9)$$

Here, $(\cdot)_{d'}$ denotes restricting the latent code to the $d'$-dimensional shared part. $f_{\phi_s}$ is an invertible neural network with affine coupling layers (Dinh et al., 2015), i.e. $f_{\phi_s} = f_{\phi_s}^K \circ f_{\phi_s}^{K-1} \circ \cdots \circ f_{\phi_s}^1$. This makes it easy to compute Jacobians $J_{\phi_s}^i = \partial f_{\phi_s}^i/\partial f_{\phi_s}^{i-1}$ formulated as triangular matrices for the layers of the invertible neural network. Again, we assume that $f_{\phi_s}^0 = I$ is the identity.

The number of dimensions for domain-specific information, i.e. the dimensionality of $z_t'$ and $z_v'$, can be different for the two domains, depending on the complexity of each distribution. Note that by conditioning on the dimensions with a supervision signal, we minimize the redundancy in the dimensions for unsupervised information without disentangling the dimensions. The multimodal conditional prior in $q_{\theta_2}(z_t'|z_s, x_t)$ and $q_{\theta_3}(z_v'|z_s, x_v)$ is modeled with non-volume-preserving normalizing flow models with Eq. (7) (Dinh et al., 2017), parameterized by $\phi_t$ and $\phi_v$, respectively.

The data log-likelihood terms in Eqs. (4) and (6) are defined as the (negative) reconstruction errors. Let $g_{\theta_v} : x_v \mapsto z_v$ with $\theta_v = \{\theta_1, \theta_3\}$ and $g_{\theta_t} : x_t \mapsto z_t$ with $\theta_t = \{\theta_1, \theta_2\}$ denote the image and text encoders, respectively. Further, $h_{\omega_v} : z_v \mapsto x_v$ and $h_{\omega_t} : z_t \mapsto x_t$ are the decoders with parameters $\omega_v$ and $\omega_t$ corresponding to images and texts, respectively, with $\tilde{x}_v = h_{\omega_v}(g_{\theta_v}(x_v))$ and $\tilde{x}_t = h_{\omega_t}(g_{\theta_t}(x_t))$ denoting the decoded image and text samples. For texts, we consider the output probability of the $j^{\text{th}}$ word $(x_t)_j$ of the ground-truth sentence $x_t$ given the previous reconstructed words $(\tilde{x}_t)_{0:j-1}$ from the text decoder, and define the reconstruction error as $L_t^{rec}(x_t, \tilde{x}_t) = -\sum_j \log p_\mu\big((x_t)_j|(\tilde{x}_t)_{0:j-1}\big)$. For images, the reconstruction loss between the input image $x_v$ and reconstructed image $\tilde{x}_v$ from the image decoder is taken as $L_v^{rec}(x_v, \tilde{x}_v) = \|x_v - \tilde{x}_v\|$, where $\|\cdot\|$ is either the $\ell_1$ or $\ell_2$ norm. Furthermore, in Eq. (9) we define $\log q_{\theta_1}\big(f_{\phi_s}((z_v)_{d'})|x_t\big)$ as the cost of mapping $(z_v)_{d'}$ to the latent space of texts under the transformation $f_{\phi_s}$; we use the mean-squared error between the encoded text representations $(z_t)_{d'}$ and the transformed image representation $f_{\phi_s}((z_v)_{d'})$ of the paired data $(x_t, x_v)$ (see Ardizzone et al., 2019).

| Method | B-4 | B-3 | B-2 | B-1 | C | R | M | S |
|---|---|---|---|---|---|---|---|---|
| CVAE (baseline)* | 0.309 | 0.376 | 0.527 | 0.696 | 0.950 | 0.538 | 0.252 | 0.176 |
| Div-BS (Vijayakumar et al., 2018) | 0.402 | 0.555 | 0.698 | 0.846 | 1.448 | 0.666 | 0.372 | 0.290 |
| POS (Deshpande et al., 2019) | 0.550 | 0.672 | 0.787 | 0.909 | 1.661 | 0.725 | 0.409 | 0.311 |
| AG-CVAE (Wang et al., 2017) | 0.557 | 0.654 | 0.767 | 0.883 | 1.517 | 0.690 | 0.345 | 0.277 |
| Seq-CVAE (Aneja et al., 2019) | 0.575 | 0.691 | 0.803 | **0.922** | 1.695 | **0.733** | **0.410** | **0.320** |
| LNFMM-MSE (pre-trained) | **0.606** | 0.686 | 0.798 | 0.915 | 1.682 | 0.723 | 0.400 | 0.306 |
| LNFMM (pre-trained) | 0.600 | **0.695** | **0.804** | 0.917 | 1.697 | 0.729 | 0.400 | 0.311 |
| LNFMM | 0.597 | **0.695** | 0.802 | 0.920 | **1.705** | 0.729 | 0.402 | 0.316 |

Table 1. Oracle performance for captioning on the COCO dataset with different metrics

| Method | B-4 | B-3 | B-2 | B-1 | C | R | M | S |
|---|---|---|---|---|---|---|---|---|
| Div-BS (Vijayakumar et al., 2018) | **0.325** | 0.430 | 0.569 | 0.734 | 1.034 | **0.538** | **0.255** | 0.187 |
| POS (Deshpande et al., 2019) | 0.316 | 0.425 | 0.569 | 0.739 | 1.045 | 0.532 | **0.255** | **0.188** |
| AG-CVAE (Wang et al., 2017) | 0.311 | 0.417 | 0.559 | 0.732 | 1.001 | 0.528 | 0.245 | 0.179 |
| LNFMM | 0.318 | **0.433** | **0.582** | **0.747** | **1.055** | **0.538** | 0.247 | **0.188** |
| LNFMM-TXT (semi-supervised, 30% labeled) | 0.276 | 0.384 | 0.529 | 0.706 | 0.973 | 0.511 | 0.241 | 0.171 |
| LNFMM-MSE (semi-supervised, 30% labeled) | 0.277 | 0.388 | 0.531 | 0.704 | 0.910 | 0.509 | 0.231 | 0.169 |
| LNFMM (semi-supervised, 30% labeled) | 0.300 | 0.413 | 0.559 | 0.729 | 0.984 | **0.538** | 0.242 | 0.172 |

Table 2. Consensus re-ranking for captioning on the COCO dataset using CIDEr

Starting from the ELBO on the right-hand side of Eq. (6), the learned latent priors in Eq. (8), the invertible mapping from Eq. (9), and plugging in the reconstruction terms as just defined, the overall objective of our semi-supervised generative model framework to be minimized is given by

$$
\begin{aligned}
\mathcal{L}_\mu(x_t, x_v) = {} & \lambda_1 D_{\mathrm{KL}}\big(q_{\theta_1}\big(z_s|x_t, x_v\big)\,\big\|\,p_{\phi_s}(z_s)\big) + \lambda_2 D_{\mathrm{KL}}\big(q_{\theta_2}\big(z_t'|x_t, z_s\big)\,\big\|\,p_{\phi_t}(z_t'|z_s)\big) \\
& + \lambda_3 D_{\mathrm{KL}}\big(q_{\theta_3}\big(z_v'|x_v, z_s\big)\,\big\|\,p_{\phi_v}(z_v'|z_s)\big) + \lambda_4 L_t^{rec}(x_t, \tilde{x}_t) + \lambda_5 L_v^{rec}(x_v, \tilde{x}_v).
\end{aligned}
\tag{10}
$$

Here, $\mu = \{\theta, \phi_s, \phi_v, \phi_t, \omega_v, \omega_t\}$ are all parameters to be learned; $\lambda_i, i = \{1, \ldots, 5\}$ are regularization parameters. Recall that $\tilde{x}_t$ and $\tilde{x}_v$ are decoded text and image samples, respectively. We assume the prior $p_{\phi_s}(z_s)$ on the shared latent space to be uniform. The strength of this uniform prior on the shared dimensions can be controlled with the regularization parameter $\lambda_1$.

Our model allows for bi-directional many-to-many mappings. In detail, given a data point $x_v$ from the image domain with latent encoding $z_v$, we first map it to the text domain through the invertible transformation $z_s = f_{\phi_s}((z_v)_{d'})$. We can now generate diverse texts by sampling from the learned latent prior $p_{\phi_t}(z_t'|z_s)$. A similar procedure is followed for sampling images given text through the learned prior $p_{\phi_v}(z_v'|z_s)$. For conditional generation tasks, as we do not have to sample from the supervised latent space, we find a "uniform" prior $p_{\phi_s}(z_s)$ to be advantageous in practice as it loosens the constraints on the decoders. Alternatively, a more complex flow-based prior could also be used here to enable sampling of the shared semantic space. We show the effectiveness of our joint latent normalizing flow-based priors on real-world tasks, *i.e.* diverse image captioning and text-to-image synthesis.

## 4 EXPERIMENTS

To validate our method for learning many-to-many mappings to provide latent joint distributions, one of the important real-world tasks is that of image-to-text or text-to-image synthesis. To that end, we perform experiments on the COCO dataset (Lin et al., 2014). It contains 82,783 training and 40,504 validation images, each with five captions. Following Wang et al. (2016); Mao et al. (2015) for image captioning, we use 118,287 data points for training and evaluate on 1,000 test images. For text-to-image synthesis, the training set contains 82,783 images and 40,504 validation data points at test time (Reed et al., 2016; Huang et al., 2017). Architecture details can be found in the Appendix.

### 4.1 IMAGE CAPTIONING

We evaluate our approach against methods that generate diverse captions for a given image. We compare against methods based on (conditional) variational autoencoders, AG-CVAE (Wang et al.,

---
*Our implementation

| Method | Unique ↑ | Novel ↑ | mBLEU ↓ | Div-1 ↑ | Div-2 ↑ |
|--------|----------|---------|---------|---------|---------|
| Div-BS | **100** | 3421 | 0.82 | 0.20 | 0.25 |
| POS | 91.5 | 3446 | 0.67 | 0.23 | 0.33 |
| AG-CVAE | 47.4 | 3069 | 0.70 | 0.23 | 0.32 |
| Seq-CVAE | 84.2 | 4215 | 0.64 | 0.33 | 0.48 |
| LNFMM | 97.0 | **4741** | **0.60** | **0.37** | **0.51** |

Table 3. Diversity evaluation on at most the best-5 sentences after consensus re-ranking

| Method | B-1 | B-4 | CIDEr |
|--------|-----|-----|-------|
| M$^3$D-GAN | 0.652 | 0.238 | - |
| GXN | 0.571 | 0.149 | 0.611 |
| LNFMM | **0.747** | **0.315** | **1.055** |

Table 4. Comparison to the state of the art for bi-directional generation

| Image | Caption | Image | Caption |
|-------|---------|-------|---------|
| 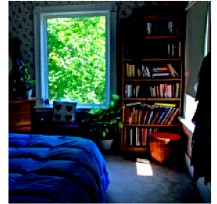 | • Two elephants standing next to each other in a river.
• A herd of elephants in a grassy area of water.
• Two elephants walking through a river while standing in the water.
• Two elephants are walking and baby in the water.
• A group of elephants walking around a watering hole. | | • A living room filled with furniture and a large window.
• A room with a couch and a large wooden table.
• A living room filled area with furniture and a couch and chair.
• A room with a couch, chair, and a lamp.
• This is a room with furniture on the floor. |

Table 5. Example captions generated by our model

2016), which uses additional supervision based on information about objects in the images for a Gaussian mixture model in the latent space, and Seq-CVAE (Aneja et al., 2019), which models sequential latent spaces with Gaussian priors. We also include Div-BS (Vijayakumar et al., 2018) based on beam search and POS (Aneja et al., 2018), which uses additional supervision from images. Additionally, we include different ablations to show the effectiveness of various components of our approach. *LNFMM-MSE* does not contain the invertible neural network $f_{\phi_s}$, *i.e.* we directly minimize the mean squared error between $(z_t)_{d'}$ and $(z_v)_{d'}$ in Eq. (10). We fix the image encodings of a VGG-16 encoder in *LNFMM (pre-trained)* for comparison to image captioning methods with pre-trained image features. Furthermore, *LNFMM-TXT* contains the latent flow for unsupervised information $f_{\phi_t}$ only for the text distribution. In this setting, we have $z_v = z_s$ and thus $z_v$ is transformed through the invertible neural network as $f_{\phi_s}(z_v)$. To show that our framework can be applied in the semi-supervised setting, we also consider limited labeled data with only 30% of the paired training data for supervision. The remaining training data is included as unpaired data for modeling domain-specific information.

**Evaluation**. We evaluate the accuracy with Bleu (B) 1-4 (Papineni et al., 2002), CIDEr (C; Vedantam et al., 2015), ROUGE (R; Lin, 2004), METEOR (M; Denkowski & Lavie, 2014), and SPICE (S; Anderson et al., 2016). For evaluating diversity, we consider the metrics of Wang et al. (2017); Aneja et al. (2019). *Uniqueness* is the percentage of unique captions generated on the test set. *Novel sentences* are the captions that were never observed in the training data. *m-Bleu-4* computes Bleu-4 for each diverse caption with respect to remaining diverse captions per image. The Bleu-4 obtained is averaged across all images. *Div-n* is the ratio of distinct *n*-grams to the total number of words generated per set of diverse captions.

**Results**. In Table 1 we show the caption evaluation metrics in the oracle setting, *i.e.* taking the maximum score for each accuracy metric over all the candidate captions. We consider 100 samples $z$, consistent with previous methods. The cVAE baseline with an image-conditioned Gaussian prior does not perform well on all metrics, showing the inability of the Gaussian prior to model meaningful latent spaces representative of the multimodal nature the underlying data distribution. The overall trend across metrics is that our LNFMM model improves the upper bound on Bleu and CIDEr while being comparable on the Rouge and Spice metrics.

Comparing the accuracy of the baseline LNFMM-MSE with LNFMM, we can conclude that learning the shared posterior distribution of $z_s$ with our invertible mapping is better than directly minimizing the mean squared error in the latent space due to differences in the complexity of the distributions. Also note that LNFMM (pre-trained) with fixed image-encoded representations has better performance compared to AG-CVAE and Seq-CVAE, in particular. This highlights that the LNFMM

| Text | Sample #1 | Sample #2 | Sample #3 | Sample #4 | Ground-truth | AttnGAN | Our LNFMM |
|---|---|---|---|---|---|---|---|
| A close up of a pizza with toppings | | | | | | | |
| A baseball player swinging a bat during a game | | | | | | | |

Figure 3. Example images generated by our LNFMM model high-lighting diversity

Figure 4. Text-conditioned samples closest to test image with IoVM

learns representations in the latent space that are representative of the underlying data distribution even for pre-trained features.

Table 2 considers a more realistic setting (as ground-truth captions are not always available) where, instead of comparing against the reference captions of the test set, reference captions for images from the training set most similar to the test image are retrieved. The generated captions are then ranked with the CIDEr score (Mao et al., 2015). While Div-BS has very good accuracy across metrics due to the wide search space, our LNFMM model gives state-of-the-art accuracy on various Bleu metrics and especially the CIDEr score, which is known to correlate well with human evaluations. More interestingly, compared to AG-CVAE with conditional Gaussian mixture priors based on object (class) information, our LNFMM model, which does not encode any additional supervised information in the latent space, outperforms the former on all accuracy metrics by a large margin. Moreover, the recent GXN (Gu et al., 2018) and $M^3DGAN$ (Ma et al., 2019) also study bi-directional synthesis with joint models in Gaussian latent spaces. Ma et al. (2019) additionally model attention in the latent space. From Table 4, we see that our method considerably outperforms the competing methods, validating the importance of complex priors in the latent space for image-text distributions. This again highlights that the complex joint distribution of images and texts captured by our LNFMM model is more representative of the ground-truth data distribution. We additionally experiment with limited labelled training data *(30% labeled)*, which shows that our approach copes well with limited paired data. We finally compare LMFMM against LNFMM-TXT to show the importance of joint learning of image and text generative models. With a generative model only for texts, the joint distribution cannot be captured effectively in the latent space.

With diversity being an important goal here, we show in Table 3 that our LNFMM method improves diversity across all metrics, with a 6.5% improvement in unique captions generated in the test set and 4741/5000 captions not previously seen in the training set. Our generated captions for a given image also show more diversity with low mutual overlap (mBLEU) compared to the state of the art. We, moreover, observe high $n$-gram diversity for the generated captions of each image. Div-BS with high accuracy has limited diversity as it can repeat the $n$-grams in different captions. POS and AG-CVAE, due to guided supervision in the latent space, offer diversity but model only syntactic or semantic diversity, respectively (Wang & Chan, 2019). Our proposed LNFMM model in Table 5 shows a range of diverse captions with different semantics and syntactic structure. Therefore, we conclude that the proposed LNFMM can effectively learn semantically meaningful joint latent representations without any additional object or text-guided supervision. The data-dependent learnt priors are thus promising for synthesizing captions with high human-correlated accuracy as well as diversity.

## 4.2 TEXT-TO-IMAGE SYNTHESIS

Given a text description, we are now interested in generating diverse images representative of the domain-specific structure of images. To that end, we include a discriminator to improve the image quality of our image decoder. Note that this does not affect the joint latent space of the LNFMM model. We evaluate our method against state-of-the-art approaches such as AttnGAN (Xu et al., 2018), HD-GAN (Zhang et al., 2018b), StackGAN (Zhang et al., 2017), and GAN-INT-CLS (Reed et al., 2016). While our main goal is to encourage text-conditioned diversity in the generated samples, the current state-of-the-art for text-to-image generation aims at improving the realism of the

generated images. Note that various GANs can be integrated with the image decoder of our framework as desired.

**Evaluation.** As we are interested in modeling diversity, we study the diversity in generated images using the Inference via Optimization (IvOM; Srivastava et al., 2017) and LPIPS (Zhang et al., 2018a) metrics against the state-of-the-art AttnGAN. Given the text, for each matching image, IvOM finds the closest image the model is capable of generating. Thus, it shows whether the model can match the diversity of the ground-truth distribution. LPIPS (Zhang et al., 2018a) evaluates diversity by computing pairwise perceptual similarities using a deep neural network. Additionally, we also report the Inception score (Salimans et al., 2016).

**Results.** In Table 6, our method improves over AttnGAN for both IvOM and LPIPS scores, showing that our method can effectively model the image semantics conditioned on the texts in the latent space, as well as generate diverse images for a given caption. Note that AttnGAN uses extra supervision to improve the inception score. However, it is unclear if this improves the visual quality of the generated images as pointed out by

| Method | IS $\uparrow$ | IvOM $\downarrow$ | LPIPS $\uparrow$ |
|---|---|---|---|
| AttnGAN | **25.89**$\pm$**0.47** | 1.101* | 0.472* |
| GAN-INT-CLS | 7.88$\pm$ 0.07 | – | – |
| Stack-GAN | 8.45$\pm$0.03 | – | – |
| HD-GAN | 11.86$\pm$0.18 | – | – |
| LNFMM | 12.10 $\pm$0.18 | **0.430** | **0.481** |

Table 6. Evaluation on text-to-image synthesis

Zhang et al. (2018b). We improve the IS over HD-GAN, which does not use additional supervision. Qualitative examples in Fig. 3 show that our LNFMM model generates diverse images, *e.g.*, close-up images of food items as well as different orientations of the baseball player in the field. In Fig. 4 we additionally see that given a caption, images generated by our LNFMM model capture detailed semantics of the test images compared to that of AttnGAN, showing the representative power of our latent space.

## 5 CONCLUSION

We present a novel and effective semi-supervised LNFMM framework for diverse bi-directional many-to-many mappings with learnt priors in the latent space, which enables modeling joint image-text distributions. Particularly, we model domain-specific information conditioned on the shared information between the two domains with normalizing flows, thus preserving the heterogeneous structure of the data in the latent space. Our extensive experiments with bi-directional synthesis show that our latent space can effectively model data-dependent priors, which enable highly accurate *and* diverse generated samples of images or texts.

**Acknowledgement.** This work has been supported by the German Research Foundation as part of the Research Training Group *Adaptive Preparation of Information from Heterogeneous Sources (AIPHES)* under grant No. GRK 1994/1.

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

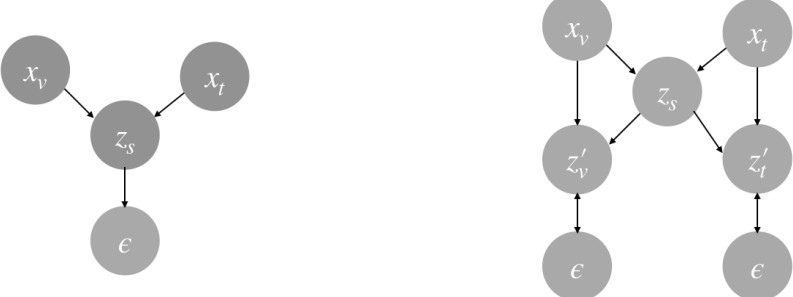

Figure 5. Inference model of our approach (*right*) in comparison to the conditional variational autoencoder of AG-CVAE (*left*; Wang et al., 2017). AG-CVAE models only supervised information in the latent dimensions. Our model encodes domain-specific variations in the conditional priors $z_v'$ and $z_t'$.

## A  APPENDIX

### A.1  NETWORK ARCHITECTURE

We provide the details of the network architecture of Fig. 2.

**Image pipeline.** The network consists of an image encoder built upon VGG-16. 4096-dimensional activations of input images are extracted from the fully-connected layer of VGG-16. This is followed by a 2048-dimensional fully-connected layer with ReLU activations. We then project it to the latent space with a 1056-dimensional fully-connected layer. The image pipeline has $d' = \dim z_s = 992$ and $\dim z_v' = 64$ dimensions. In the image decoder, we leverage the architecture of Zhang et al. (2019a) to synthesize images of $64 \times 64$ or $256 \times 256$ dimensions. The input to the decoder has dimensionality 1056. For the image generation experiments, we additionally apply the discriminator of Zhang et al. (2019b) to the output of the image decoder.

**Text pipeline.** We use a bidirectional GRU with two layers and a hidden size of 1024 as text encoder. This outputs 1024-dimensional latent representations for sentences. For text, $\dim z_t' = 32$. The text decoder is a LSTM with one layer and hidden size of 512.

**Flow modules.** Our network consists of two flow modules for conditional priors on image and text domains and an invertible neural network to exchange supervised information.

*Invertible neural network for supervision ($f_{\phi_s}$):* This invertible neural network consists of 12 flow layers and input dimensionality of 992. Each flow consists of conditional affine coupling layers followed by a switch layer (Dinh et al., 2015).

*Latent flow for conditional prior on images ($f_{\phi_v}$):* We map the 64 dimensions of the image encodings from the image encoder to a Gaussian with normalizing flows with 16 layers of flow and 512 hidden channels. Each flow consists of conditional affine coupling layers followed by a switch layer (Dinh et al., 2017).

*Latent flow for conditional prior on texts ($f_{\phi_t}$):* We map the 32 dimensions of the text encodings from the text encoder to a Gaussian with normalizing flows with 16 layers of flow and 1024 hidden channels. Each flow consists of a conditional activation normalization layer followed by conditional affine coupling layers. Invertible $1 \times 1$ convolutions are applied to the output of the affine coupling layers, which is followed by a switch layer (Kingma & Dhariwal, 2018).

### A.2  DIVERSITY IN IMAGE CAPTIONING

We show more qualitative examples of the captions generated by our LNFMM model in Table 7. The example captions show syntactic as well as semantic diversity.

| Image | Caption | Image | Caption |
|---|---|---|---|
| | • A woman holding an umbrella while standing in the rain.
• A woman is holding umbrella on the street
• A woman walking a street with a umbrella in the rain.
• A woman is holding an umbrella while walking in the rain
• A woman walking down a street while holding an umbrella | | • A woman standing in front of a refrigerator
• Two people standing together in a large kitchen
• Two people are standing in a kitchen counter.
• A family is preparing food in a kitchen.
• A few people standing in the kitchen at a table. |
| | • A man is holding a tennis racket on the tennis court.
• A tennis player about to hit a tennis ball
• A man is playing tennis with a racket on the tennis court.
• A man standing on a tennis court is holding a racket
• A man prepares to hit a ball with tennis racket. | | • A large clock tower is in the middle of a building.
• A tall building with a clock tower in front of a building.
• A clock tower in the sky with a clock on top.
• A tall building with a clock on top.
• A tall clock tower with a clock tower on it. |

Table 7. Example captions generated by our LNFMM model

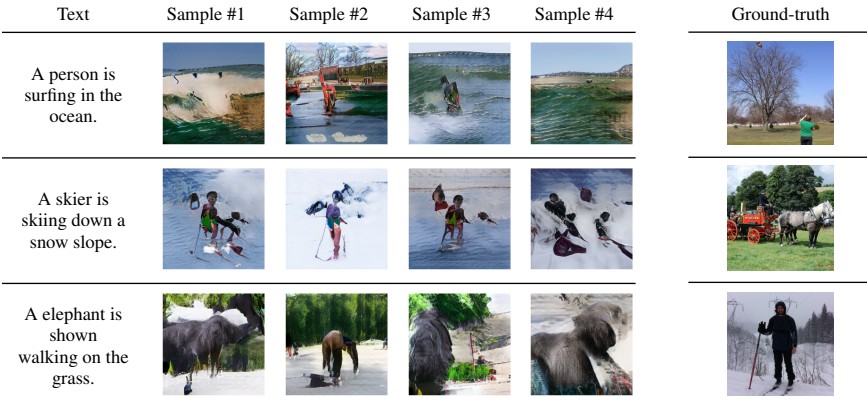

Figure 6. Example images generated by our LNFMM model highlighting diversity

Figure 7. Text-conditioned samples closest to test image with IvOM by our LNFMM

## A.3 TEXT-TO-IMAGE SYNTHESIS

We additionally show diverse images generated by our LNFMM model in Fig. 6. Our generated images can successfully capture the text semantics and also exhibit image specific diversity, *e.g.* in style and orientation of objects. Furthermore, to show that the latent space captures the joint distribution, we show the images generated by our model with IvOM by finding a $z_v$ conditioned on the input text that is most likely to have generated the test image. We show the images generated for the $z_v$ most likely to have generated the image. Our generated samples in Fig. 7 capture the details in the images, showing that our LNFMM model learns powerful latent representations.

## A.4 VISUALIZATION OF LATENT SPACES

In Table 8 we show the t-SNE (van der Maaten & Hinton, 2008) visualization of the latent space of texts conditioned on an image. In the examples shown, we observe global patterns like 'table with foods' across all captions of an image and local patterns in clusters like 'table with several' or 'a dinner table with' showing image-conditioned representation as well as domain-specific variation.

| Image | Latent Space | Captions |
|---|---|---|

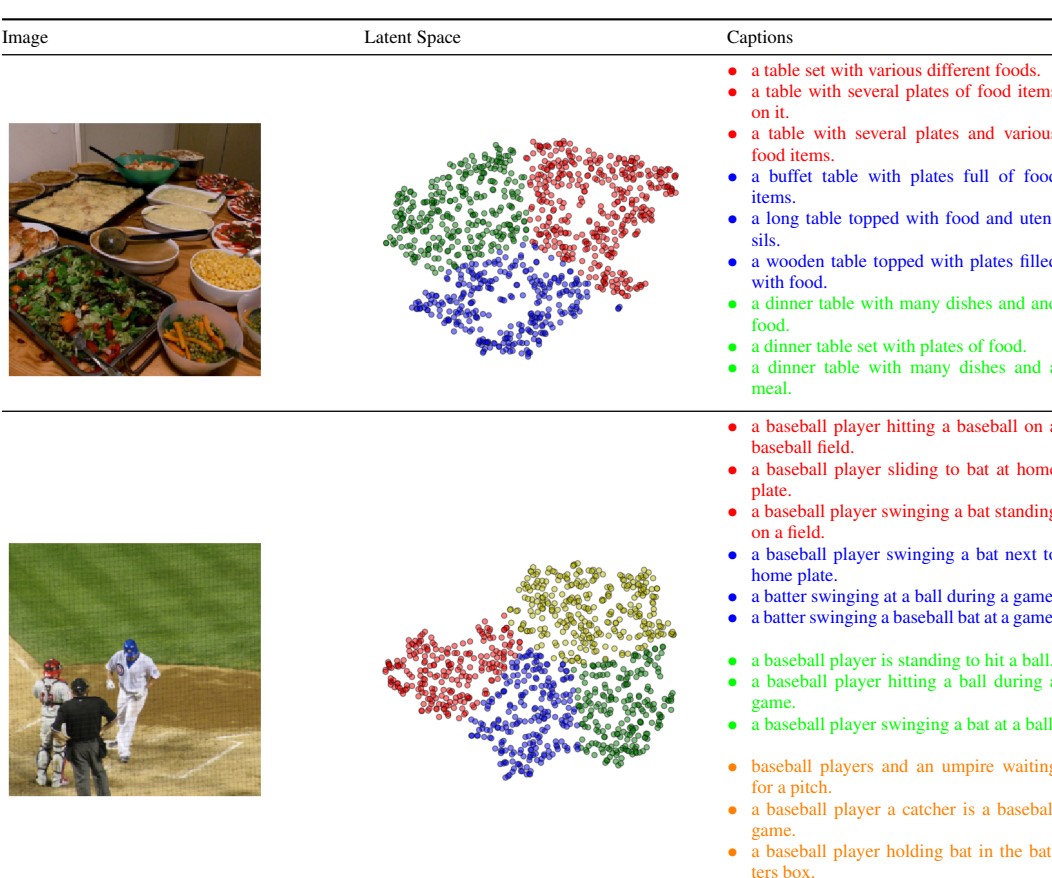

Table 8. t-SNE visualization of the latent space of texts conditioned on an image. Example captions clustered using $k$-means. Captions are color-coded based on corresponding clusters

Similarly, in Table 9 we can observe clusters in image space conditioned on texts. Here again, we can observe global patterns like 'close-up of food' and local patterns like 'pizza', 'food with toppings, or 'sandwiches'.

## A.5 VALIDATION WITH HUMAN EVALUATION

In addition to validating the performance of our LNFMM method on image captioning for accuracy in Table 1 and for diversity Table 3 with various measures, we conduct a human evaluation of the quality of captions generated by our LNFMM approach against Div-BS (Vijayakumar et al., 2018) for accuracy and diversity. We presented five captions each for a set of images and for each method to four human annotators. The human annotators assessed the captions for accuracy and diversity on a scale from 1 to 10. Here, accuracy is defined as how well the captions describe the details in a given image and diversity can be syntactic diversity or semantic diversity.

In Fig. 8 (*left*), the score for each image is the average score given by the four annotators and Fig. 8 (*right*) shows the scores for the captions of each image from each annotator. Here, our findings are similar to those in Table 1 and Table 3. The captions generated by Div-BS were assessed to have diversity scores in the range of $[5, 8]$, while the diversity scores fell into the range $[7, 10]$ for our LNFMM method. Furthermore, the accuracy of the captions of each image generated with our LNFMM was found to be better compared to Div-BS, thereby showing that LNFMM can generate captions that are both diverse and accurate.

| Text | Latent Space | Generated Images |
|------|--------------|------------------|

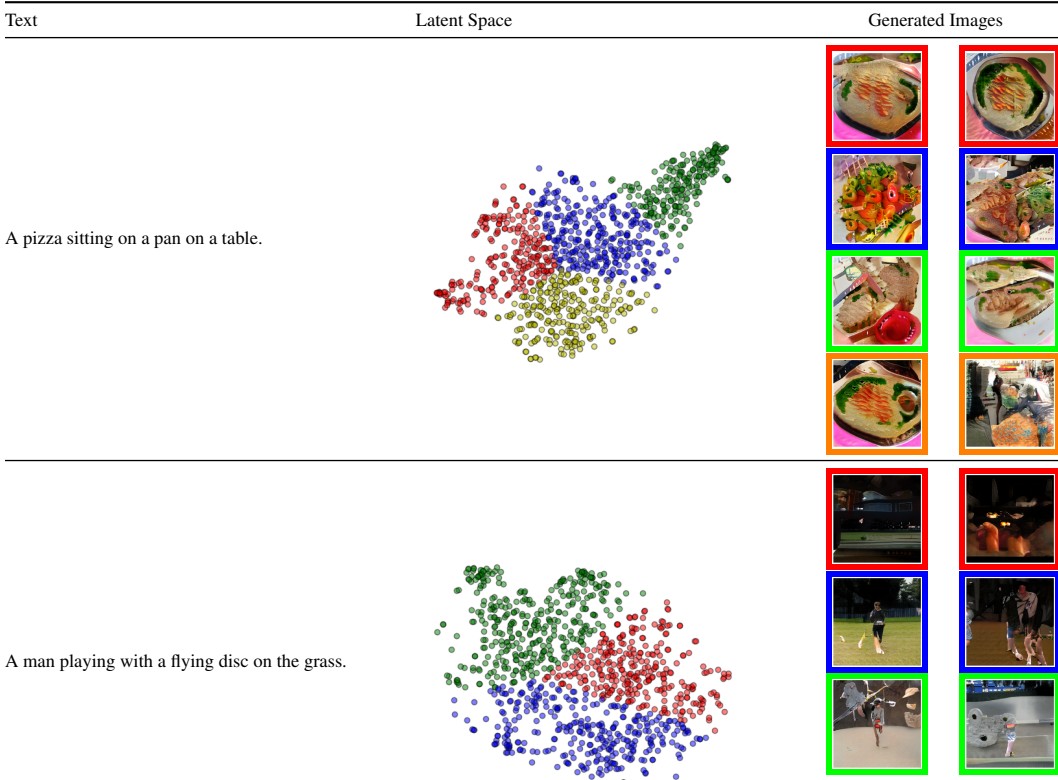

A pizza sitting on a pan on a table.

A man playing with a flying disc on the grass.

Table 9. t-SNE visualization of the latent space of images conditioned on a caption. Example images clustered using $k$-means. Images are color-coded based on corresponding clusters

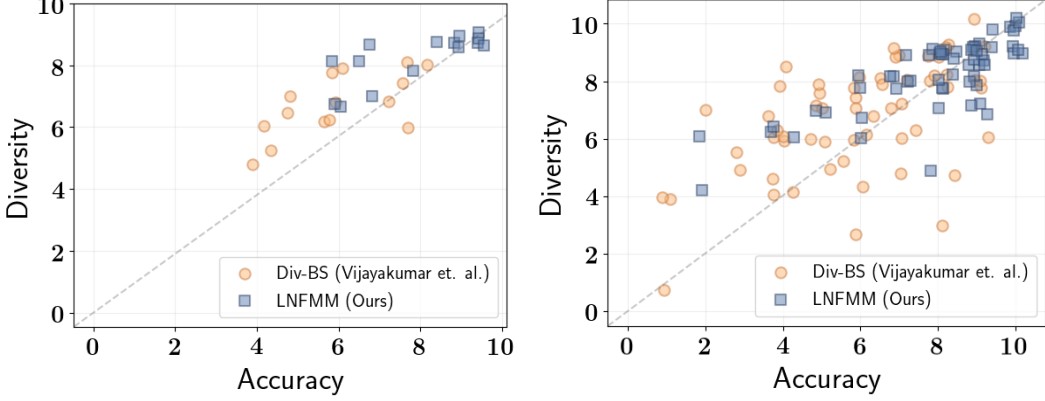

Figure 8. Comparison of accuracy and diversity scores of captions for images generated with LNFMM (Ours) and DIV-BS (Vijayakumar et al., 2018): Scores for captions of an image averaged across all annotators (*left*) and scores for all captions of an image for each annotator (*right*). To make coinciding scores be easier to see, a small random jitter is added to each human assessment.

