# OpenReview forum: "Latent Normalizing Flows for Many-to-Many Cross-Domain Mappings"
_ICLR.cc/2020/Conference — Accept (Poster)_

### Official Review · AnonReviewer2 · 2019-10-23
**Official Blind Review #2**

**Rating:** 8

**Review:**

Summary:
This paper addresses the problem of many-to-many cross domain mapping tasks (such as captioning or text-to-image synthesis). It proposes a double variational auto-encoder architecture mapping data to a factored latent representation with both shared and domain-specific components. The proposed model makes use of normalizing flow-based priors to enrich the latent representation and of an invertible network for ensuring the consistency of the shared component across the two autoencoders. Experiments are thorough and demonstrate results that are competitive or better than the state-of-the-art.

Decision:
This work is a good example of meticulous and well-executed neural network engineering. It combines well-known ideas (variational auto-encoders, normalizing flow priors, invertible networks) into an effective and working solution for a complicated problem. The model is shown to bring improvements in the state-of-the-art over several metrics and benchmarks. The manuscript is well written and easy to follow (provided some technical familiarity with variational inference). It includes all the necessary details for understanding the method. Related works appear to have been discussed and compared properly, although I cannot assess if important works on cross-domain mapping are missing. For these reasons, I recommend this work for acceptance without reservation.

Additional feedback:
- Above Eqn 5: K- divergence --> KL divergence
- The code could have been cleaned up and better organized, for easier reproducibility and reuse.

---

Post-rebuttal update: Thank you for your answers. I still recommend your work for acceptance.

**Experience Assessment:**

I have read many papers in this area.

**Review Assessment: Checking Correctness Of Derivations And Theory:**

I assessed the sensibility of the derivations and theory.

**Review Assessment: Checking Correctness Of Experiments:**

I assessed the sensibility of the experiments.

**Review Assessment: Thoroughness In Paper Reading:**

I read the paper at least twice and used my best judgement in assessing the paper.

---

> ### Author Response · Authors · 2019-11-11
> **Response to Review #2**
>
> Thanks a lot for your very encouraging feedback. We really appreciate it and wanted to give brief responses to your two concerns.
>
> * Above Eqn 5: K- divergence --> KL divergence:
> Thank you. We have fixed this.
>
> * The code could have been cleaned up and better organized, for easier reproducibility and reuse:
> Thank you for the comment. We have already made some improvements to aid re-use. We will make a complete, cleaned version available soon.
>
> We would be happy to answer any additional questions.

---

### Official Review · AnonReviewer1 · 2019-10-26
**Official Blind Review #1**

**Rating:** 8

**Review:**

Summary:

The paper proposes a model for joint image-text representations

The paper proposes a model fro cross-domain generative tasks, specifically image captioning and text-to-image synthesis. The proposed Latent Normalizing Flows for Many-to-Many Mappings uses normalizing flows to model complex joint distributions. The latent representation consist of domain-specific representation and cross-domain information shared across image and text using invertible metrics.

Novelty:

- The paper explores an interesting area of learning representations for cross-domain tasks such as image captioning and text-to-image synthesis.
- The model is well explained.  Section 3 explains the model formulation nicely, has consistent notation and slowly  builds up to the final formulation by explaining each component concisely.
- Recent methods like VQ-VAE [1]  have shown promising results for image generation. The related work doesn't provide any discussion regarding that.

Experiments / Analysis:

- The model contains exhaustive experiments for both image-captioning and image generation. The model performs on-par or beat state of the art methods on both perceptual and diversity metrics. On diversity metrics, the model performs much better than other recent methods like Seq-CVAE (which arrived on ArXiv only a few weeks prior to the submission deadline) and POS.
- The model also shows results on text-to-image synthesis comparing with multiple baselines and various diversity metrics and inception score.
- While the proposed methods beats existing diversity and perceptual metrics, it'd be good to also run a human study since these metrics are only a proxy to human judgement.
- Apart from showing empirical result, the paper can benefit from providing Insights what the domain specific representation has learnt and what the cross-domain representation learnt.
- Can the model benefit from training on unaligned image and textual data to learn better domain specific representations?

Clarity:
- Ablations not clear: It's not a 100% clear from the paper what LNFMM-MSE and LNFMM-TXT mean. "LNFMM-TXT contains unsupervised dimensions only for the text distribution and
all encoded image features are used for supervision, i.e.without fφv" What does this sentence mean?  Similarly, it's not clear what "LNFMM (semi-supervised, 30% labeled)" mean?
- It's also not clear why the authors call the approach a semi supervised setup? For instance,  the paper relies on supervision from  paired image-caption data to train the model.

[1] Neural Discrete Representation Learning; Aaron van den Oord, Oriol Vinyals, Koray Kavukcuoglu


**Update after rebuttal**
Thank you for clarifications to my questions regarding the ablations and  using unaligned image and textual data to learn better domain specific representations. The visualizations in Appendix A.4 are also somewhat helpful in understanding what the representations have learnt. After reading the rebuttal, and considering that they will run the human study to further validate their approach in the final manuscript, I am happy to raise the score.

**Experience Assessment:**

I have published one or two papers in this area.

**Review Assessment: Checking Correctness Of Derivations And Theory:**

I assessed the sensibility of the derivations and theory.

**Review Assessment: Checking Correctness Of Experiments:**

I carefully checked the experiments.

**Review Assessment: Thoroughness In Paper Reading:**

I read the paper at least twice and used my best judgement in assessing the paper.

---

> ### Author Response · Authors · 2019-11-11
> **Response to Review #1**
>
> Thank you very much for the constructive and very detailed comments. We are glad that you appreciated novelty, clarity, and experiments and happily comment on the raised concerns.
>
> * VQ-VAE related work:
> Thank you for the pointer. We have added the reference in the related work section. We agree that VQ-VAE shows high quality image generation results. In contrast to our LNFMM framework and other prior work, e.g. Seq-CVAE (Aneja et al. 2019), M3D-GAN (Ma et al. 2019), which models images and/or text as continuous distributions in the latent space, the VQ-VAE framework relies on discrete latent variables. Extending this to joint distributions, e.g. images and texts, is certainly an interesting direction for future work.
>
> * Human study:
> Thank you for the suggestion. We agree that existing metrics are only a proxy to human judgement. We will definitely consider including a human study when extending the manuscript.
>
> * Insights what the domain specific representation has learnt and what the cross-domain representation learnt:
> We have included visualizations for domain-specific and cross-domain information in the Appendix A.4 (Visualization of Latent Spaces) of the revised paper.
>
> * Can the model benefit from training on unaligned image and textual data to learn better domain specific representations?
> Unaligned image and text data can be included during training for better domain-specific representations. The closest experiments in this setting are in Tab. 2, where we include only 30% of the training data as aligned image-text pairs and the remaining 70% are included as unaligned image & text data. We observe that LNFMM (semi-supervised, 30% labeled) with domain-specific components for both images and texts performs better than the baseline LNFMM-TXT (semi-supervised, 30% labeled) for which the domain-specific component is present only for the texts. This shows that the model can benefit from the domain-specific components of each domain. Furthermore, comparing the results of LNFMM with LNFMM (semi-supervised, 30% labeled) we observe that the performance of the model does not drop considerably given limited amount of paired data for supervision. Thus the model benefits from the unaligned data of images and texts for learning latent representations with good performance on various evaluation metrics in Tab. 2.
>
> * Ablations not clear:
> Our main contributions are the domain-specific multimodal priors for images $(p_{\phi_v})$ and texts $(p_{\phi_t})$ for modeling domain-specific information in the latent space and an invertible neural network $f_{\phi_s}$ for transforming data points from one domain to the other. Our complete model is denoted by LNFMM and we show the benefits of the different components by including the following ablations:
>
> - A CVAE baseline is included to show the advantage of learning multimodal priors in latent space over a standard Gaussian prior, which cannot capture the multimodality of the data in the latent space.
> - We include LNFMM-MSE, where we perform supervision by directly minimizing the mean squared error along the shared d' dimensions of text and image encodings, i.e. $||(z_t)_d'-(z_v)_d'||^2$. We remove the invertible neural network $f_{\phi_s}$ for this ablation. We include this baseline to show the effect of the invertible neural network $f_{\phi_s}$ for aligning the latent representations of images and texts for cross-modal tasks (e.g. image captioning).
> -In LNFMM-TXT, we remove the domain-specific information component from the image pipeline, i.e. $p_{\phi_v}$. Here, we have $z_v = z_s$. The domain-specific component is included only for texts, i.e. $p_{\phi_t}$. This ablation is included to show the benefits/effects of domain-specific components for learning latent representations on cross-modal tasks (e.g. image captioning).
> - LNFMM (semi-supervised, 30% labeled) includes the results of our model in a semi-supervised setting, in which we used 30% of the training data for supervision, i.e. image-caption pairs for 30% of the training data. The remaining images and captions are included in an unpaired fashion.
>
> We have updated the manuscript to make these ablation settings clearer.
>
> * It's also not clear why the authors call the approach a semi supervised setup:
> The approach is semi-supervised since unpaired images and texts can also be included during training in addition to paired images and captions (texts). We show the results for the semi-supervised setup in Tab. 2, where from the COCO dataset we used 30% of the training data for supervision (through pairs) and the remaining data is included as unpaired (unaligned) images and texts (see also Reviewer 3).
>
> We would be happy to answer any additional questions.

---

### Official Review · AnonReviewer3 · 2019-10-27
**Official Blind Review #3**

**Rating:** 6

**Review:**

The paper introduces a variational model for text to image and image to text mappings. The novelty consists in separating the modeling of text and image latent representations on one hand and the modeling of a shared content representation on the other hand. Priors for text, image and shared representations are generated through an invertible – flow model. The motivation for this is to allow for complex priors. Training for the shared component is supervised using aligned text and image data, while training for the residual text and image components is unsupervised. Experiments are performed for text and image generation, using training data from the COCO dataset.
The proposed model presents several innovations: separate unsupervised modeling of text and image and joint supervised modeling of shared latent variables, the use of three normalizing flows for the priors respectively associated to these variables. The intuition behind the model is well introduced. However, the technical description of the model itself is somewhat imprecise. Particularly section 3.3 describing the shared component and the global model should be carefully checked. Both descriptions are too imprecise e.g. the d’ dimensional component of z_v, z_t are not introduced; the derivation or explanation of eq. (10)  is not provided, J_phi in eq (9) not defined, etc. There are some typos or erros, check eq (7), (8), q_phi I instead of q_theta in §3.3.
The experiments compare the model with several different baselines and are quite extensive. Please indicate whether you performed all the tests yourself or picked the numbers in the literature. A better description of the baselines characteristics and of the model variants (MSE, TXT) and  their relations with the proposed model, in this paragraph, would help appreciate the results.
The proposed model seems to compare well with different baselines, but the presentation of the experiments is not that clear. For example, the ablation study in Table 1 basically shows that the Phi_s flow component behaves similarly to the complete flow model. This MSE variant is not used anymore in the other comparisons, why? Same remark for the other baselines, why some are used and some not in the different tests? The same remark hold for the text to image experiments.
 Overall, there are interesting new ideas, a new model, insufficient model description and experiments details.


----- After rebuttal -----------

The authors made an effort to clarify and correct the technical errors. I still have some concerns with confusing notations.  But I keep my score.


**Experience Assessment:**

I have read many papers in this area.

**Review Assessment: Checking Correctness Of Derivations And Theory:**

I assessed the sensibility of the derivations and theory.

**Review Assessment: Checking Correctness Of Experiments:**

I assessed the sensibility of the experiments.

**Review Assessment: Thoroughness In Paper Reading:**

I read the paper thoroughly.

---

> ### Author Response · Authors · 2019-11-11
> **Response to Review #3**
>
> Thank you very much for the constructive feedback and the concrete suggestions for improving the manuscript. We very much appreciate that you found our work innovative with interesting new ideas, and commented on the novelty of our approach. We next address the raised concerns in detail.
>
> * Shared component and the global model should be carefully checked:
> Thank you for pointing out these typos/imprecisions in the formalization. We have made a careful pass of Sec. 3 in the revised paper and addressed these issues. We believe the notation to be significantly improved.
>
> * A better description of the baselines characteristics and of the model variants (MSE, TXT):
> We have added indications in the manuscript to denote that we implemented the baseline (CVAE) ourselves. Other results are taken directly from the literature (previous state-of-the-art approaches, i.e. DIV-BS, AG-CVAE, POS, Seq-CVAE).
> We also updated the manuscript with more detailed descriptions of the baselines and our the model variants (see also comments to Review #1).
>
> * MSE variant is not used anymore in the other comparisons:
> In Tab. 1 we include the MSE baseline when we report the oracle performance of different baselines and the state of the art regarding different performance metrics. For accuracy, in line with previous work, we take our model with the highest CIDEr score (LNFMM) and evaluate it for consensus re-ranking (Tab. 2) and diversity metrics (Tab. 3). We used the same best-performing model for text-to-image generation experiments.
> For completeness, we have included the MSE baseline for the experiments with limited labeled data (Tab. 2), i.e. LNFMM-MSE (semi-supervised, 30% labeled). We again observe a considerable advantage of our complete LNFMM model over LNFMM-MSE, showing the advantage of our supervised flow component $f_{\phi_s}$.
>
> We would be happy to answer any additional questions.

---

> > ### Comment · AnonReviewer3 · 2019-11-15
> > **comments on your answer**
> >
> > Thanks for your corrections and answers.
> > I checked the new version and apparently there are still errors in the formulas.
> > Eq. (7). 2nd line is wrong. Your notation  for the Jacobians is not correct or at least confusing, as such this is not the chain rule. Then if I understand correctly there is a sign error - instead of + in the 2nd row). It would help the reader to write f the other way around, f=f^Ko...of^1.
> > Eq (8) sign error
> > Eq. (9) wrong
> >
> > section 3.3 remains unclear for me.

---

> > > ### Author Response · Authors · 2019-11-15
> > > **Response to Reviewer #3**
> > >
> > > Thank you for your careful check!
> > >
> > > * Eq. 7: Following your suggestion, we have written the concatenation of the $f_i$ the other way around, to also be consistent with previous work such as Glow. This implies the sign change you mentioned.
> > >
> > > * Eq. 8: Yes, thank you.
> > >
> > > * Eq. 9: We fixed the typos. Thank you.
> > >
> > > * Sec. 3.3: We have tried to make the derivation of the objective clearer. If you point out specific steps that remain unclear, we are happy to clarify these in the final version.

---

### Public Comment · ~Rahul_Mehta1 · 2019-10-29
**Code Reproduction**

I am unable to follow parts of the code and reproduce the results shared in the paper.

In the main training file how are the lambda values chosen ?  Currently they are set as 20, 0.6 (lambda2), 25, 500, and 1.2 .

In the function autoencode_image, why is the reconstruction computed as a function of the negative log likelihood (rec_loss + 10*nll) .

In latent_align_modules the shared dimension is modeled as a conditional affine coupling layer, what is the architecture of this layer ?  If this follows RealNVP, usually the scale is a function of MLP and the input, but in this case it is an individual parameter.

---

> ### Author Response · Authors · 2019-11-01
> **Clarifications Provided**
>
> Thankyou for your interest in our work.
>
> *Choice of lambda values*
> The regularization parameters were chosen to maximise accuracy on the validation set (4000 points)  of the MSCOCO dataset consistent with previous work. We identified a parameter range by grid search and then performed random sampling in that range.
>
> *Autoencode_image*
> In Autoencode_image we return the reconstruction and the loss. However, that particular loss was a left over from debugging.  However, we have since cleaned the code and a newer version is available at the same link with better readability. Please note, that the reconstruction is not a function of the negative log likelihood.
>
> *Affine coupling layer of latent affine module*
> We use affine coupling layers based on NICE (Dinh et al. 2015) where the scale parameter is not conditioned on the input. Please note that there is a typo in the Appendix (we cited 2017 instead of 2015 paper), we will fix this.
>
> Please note that we could reproduce our results from the publically available code.

---

> > ### Public Comment · ~Rahul_Mehta1 · 2019-11-05
> > **Code Reproduction**
> >
> > Thanks for the responses!
> >
> > Just to clarify I am able to use your code to reproduce the results, just not if I follow the paper itself.  Can you please clarify why you update the Latent Parameters every other iteration (if that is happening in the training code)?  You update the discriminator at an odd iteration and at an even iteration you update your model.
> >
> > For the affine architecture, in NICE the hidden layers are a function of the partition of the input (x1 and x2), but yours is a function of the hidden layers along with the condition.  Can you please explain why it is necessary to concatenate the condition at every layer?  (lines 135-148 , latent align modules). Based on other conditioning based VAEs many of them concatenate the condition only on the first layer.

---

### Decision · Program_Chairs · 2019-12-19

**Decision:**

Accept (Poster)

**Comment:**

This paper addresses the problem of many-to-many cross-domain mapping tasks with a double variational auto-encoder architecture, making use of the normalizing flow-based priors.

Reviewers and AC unanimously agree that it is a well written paper with a solid approach to a complicated real problem supported by good experimental results. There are still some concerns with confusing notations, and with human study to further validate their approach, which should be addressed in a future version.

I recommend acceptance.